

# Shape familiarity modulates preference for curvature in drawings of common-use objects

Erick G. Chuquichambi[1], Letizia Palumbo[2], Carlos Rey[1] and Enric Munar[1]

[1] Human Evolution and Cognition Group (EvoCog), University of the Balearic Islands, Palma, Balearic Islands, Spain
[2] Department of Psychology, Liverpool Hope University, Liverpool, United Kingdom

## ABSTRACT

Drawing is a way to represent common-use objects. The contour of an object is a salient feature that defines its identity. Preference for a contour (curved or angular) may depend on how familiar the resulting shape looks for that given object. In this research, we examined the influence of shape familiarity on preference for curved or sharp-angled drawings of common-use objects. We also examined the possibility that some individual differences modulated this preference. Preference for curvature was assessed with a liking rating task (Experiment 1) and with a two-alternative forced-choice task simulating approach/avoidance responses (Experiment 2). Shape familiarity was assessed with a familiarity selection task where participants selected the most familiar shape between the curved and the angular version for each object, or whether both shapes were equally familiar for the object. We found a consistent preference for curvature in both experiments. This preference increased when the objects with a curved shape were selected as the most familiar ones. We also found preference for curvature when participants selected the shape of objects as equally familiar. However, there was no preference for curvature or preference for angularity when participants selected the sharp-angled shapes as the most familiar ones. In Experiment 2, holistic and affective types of intuition predicted higher preference for curvature. Conversely, participants with higher scores in the unconventionality facet showed less preference for the curved drawings. We conclude that shape familiarity and individual characteristics modulate preference for curvature.

# INTRODUCTION

Common-use objects are perceived as utilitarian, familiar and hedonic products (*Wang, Yu & Li, 2019*). These characteristics influence how we interact with them daily. For instance, utility, familiarity and/or hedonism might be factors that contribute to generally preferring common-use objects with curved contours over sharp-angled ones (*Bar & Neta, 2006*; *Bar & Neta, 2007*; *Munar et al., 2015*). Preference for curvature was shown using drawings of car interiors (*Leder & Carbon, 2005*), pictures of windows (*Naghibi Rad et al., 2019*), furniture (*Dazkir & Read, 2012*), product packaging (*Westerman et al., 2012*), exterior

Corresponding author
Erick G. Chuquichambi,
erik.chuquichambi@uib.es

façades (*Ruta et al., 2019*) and interior architectural environments (*Van Oel & Van den Berkhof, 2013*; *Vartanian et al., 2013*; *Vartanian et al., 2017*), among others. While most of these stimuli involve representational content, preference for curvature was also found using non-representational art-related stimuli such as abstract artworks (*Ruta et al., 2021*) or abstract shapes and patterns (*Bertamini et al., 2016*; *Bertamini et al., 2019*).

Previous studies suggested that shared preferences are more usual with representational stimuli than abstract stimuli (*Vessel & Rubin, 2010*; *Schepman, Rodway & Pullen, 2015*; *Schepman et al., 2015*). *Rodway et al. (2016)* proposed that liking for representational stimuli is influenced by associations developed with the subject matter or semantic content of the picture. Therefore, our experience with the representational content of drawings or with the way an object is represented might also make preference for these stimuli more systematic and predictable. Skilled artists design representational drawings with relative ease (*Kozbelt et al., 2010*). On the one hand, the design process involves decisions about proportions, shading, lines, or colors, among others. On the other hand, the design process also involves implicit constraints such as the objects' functionality and usability, and sometimes even the cost of production (*Lawson, 1980*; *Kavakli et al., 1999*; *Bertamini & Sinico, 2019*).

## Preference for curvature and familiarity

The consistency of visual preference for the representational content of stimuli highlights its association with familiarity (*Reber, Winkielman & Schwarz, 1998*; *Reber, Wurtz & Zimmermann, 2004*). *Berlyne (1971)* considered that familiarity strongly influences the psychobiological mechanisms underlying aesthetic experiences. Therefore, increased exposure to specific visual features might also modulate the potential preference for the same visual features. In this regard, some studies suggested that curved contours are more frequent in natural scenes than sharp-angled ones (*Koenderink, 1984*; *Hoffman & Singh, 1997*). *Ruta et al. (2019)* used a dynamic computational model of the visual cortex and a model that characterizes discomfort in terms of adherence to the statistics of natural images (*Penacchio, Otazu & Dempere-Marco, 2013*; *Penacchio & Wilkins, 2015*) to analyze the statistical properties of drawings of architectural façades with different contour types (curved, mixed, sharp-angled and rectilinear). They found that stimulus preference was related in both models and it matched the behavioural findings of preference for façades. Therefore, they suggested that the link between the statistical properties of natural scenes and preference for curvature might have evolved from human interaction with natural environments. Other studies suggested a faster speed of processing smooth contours over angular ones (*Bertamini, Palumbo & Redies, 2019*; *Chuquichambi et al., 2020*). *Bertamini, Palumbo & Redies (2019)* argued that this advantage may be explained because curved features tend to match the statistics of the natural environment in which the visual system has evolved. However, preference for curvature may also be a context-specific effect, and not extend to all natural environment stimuli (*Hůla & Flegr, 2016*).

The influence of familiarity on preference might be also related to the proximity of an object to the category prototype. In general, we would expect a link between typicality and familiarity (*Hekkert, Snelders & Wieringen, 2003*). *Whitfield & Slatter (1979)*

investigated whether proximity to the prototype influenced aesthetic choice using images of furniture with different styles. These authors found that the furniture and styles selected by participants in their similarity task consistently corresponded to those selected in their aesthetic task. Influenced by these results, *Whitfield & Slatter (1979)* developed the preference for prototypes theory suggesting that aesthetic choice reflects categorization and prototypicality. That is, prototypicality may act as an influential determinant of preference for everyday objects (*Whitfield, 1983*). *Winkielman et al. (2006)* proposed that part of preference for prototypicality arises from a general mechanism linking fluency and positive affect. Along with prototypicality, these authors suggested that other factors might also act as fluency-enhancing variables and, therefore, explain the prototypicality-attractiveness relationship. In this sense, preference for objects with curved contours might be one of these variables because curvature facilitates processing fluency (*Corradi & Munar, 2020*).

Drawings of common-use objects are characterised by meaningful and familiar content (*Hekkert, Snelders & Dirk, 1995*; *Hekkert, Snelders & Wieringen, 2003*). They involve the perceiver's previous knowledge and momentary perceptual experience (*Leder et al., 2004*). Given that people might be more exposed to curved contours than to sharp-angled ones in daily life, the potential preference for curved drawings of common-use objects might be modulated or explained by the degree of familiarity of these objects. However, this relationship might be also modulated by the artistic reproduction of drawings.

## Drawings as artistic works

Drawings are associated with innovation and creativity because of their art-related nature (*Purcell & Gero, 1998*). The experience of drawing embodies abstract and high-level design ideas, and allows some degree of uncertainty about how to represent the physical attributes of the object (*Gross et al., 1988*). These characteristics might differentiate preference for representational drawings from preference for more realistic (e.g., photographs) or more abstract stimuli (e.g., irregular polygons). Contrary to representational stimuli, *Bornstein (1989)* found that abstract paintings, drawings, and matrices did not show a strong mere exposure effect. This effect proposes that affect increases with repeated unreinforced exposure of a stimulus, and therefore, familiarity (*Zajonc, 1968*). *Leder (2001)* also showed that repeated exposure had little effect on art-related stimuli. Instead, he suggested that familiarity-liking relations were weakened by knowledge and were greater in spontaneous judgements. These findings are compatible with the fact that novelty is an important factor in the appreciation of fine arts, where the seeking for novelty is a dominant force in its development (*Martindale, 1990*). *Hekkert, Snelders & Wieringen (2003)* showed typicality and novelty as equally effective predictors to explain aesthetic preference of consumer products (e.g., telephones, cars, etc.). They suggested that there should be a balance between novelty and typicality in the design of common-use objects. Interestingly, *Park, Shimojo & Shimojo (2010)* found segregation of preference across objects' categories, with familiarity dominant in faces, and novelty dominant in natural scenes. Given this context, the interaction between the representational content and art-related characteristics of drawings of common-use objects might contribute to understanding the role of familiarity in predicting aesthetic judgements (*Sluckin, Hargreaves & Colman, 1982*).

## Individual differences and preference for curvature

Individual differences also modulate aesthetic judgements (*Child, 1962*; *Child, 1965*; *Leder et al., 2019*). However, the influence of individual differences in preference for curvature diverges between studies. *Silvia & Barona (2009)* investigated the role of artistic expertise in preference for curvature using arrays of circles and hexagons, and asymmetrical random polygons. Although they found an interaction between art training with angular stimuli, this interaction changed depending on the specific stimuli set. *Vartanian et al. (2017)* also found divergent results in preference for curvature among experts (architects or designers) and non-experts. They presented these participants with images of curvilinear and rectilinear architectural interior spaces in a beauty judgement task and an approach-avoidance decision task. Despite that the experts found curvilinear spaces more beautiful than rectilinear ones, contour did not affect their willingness to enter or exit these spaces. Conversely, contour had no effect on judgements of beauty among nonexperts, but they were more likely to enter curvilinear spaces than rectilinear ones. However, a more recent study did not confirm preference for curved interior spaces with quasi-experts in industrial design (*Palumbo et al., 2020*), hence highlighting that individual differences might also depend on the specific training received in the area of expertise. *Cotter et al. (2017)* also reported that artistic expertise, a personality trait such as openness to experience, along with other cognitive traits (i.e., holistic thinking) predicted higher preference for curvature using irregular polygons, but not using arrays of circles and hexagons. *Corradi et al. (2019a)* suggested that aesthetic sensitivity to curvature coexists with a remarkable individual variation on people's judgements. They presented real objects and abstract designs to art and non-art students in a two-alternative forced-choice task. They also were interested in the role of sex, openness to experience and artistic expertise. Both groups of students preferred the curved stimuli but none of the individual variables showed significant results.

## The present study

In this study, we examined preference for contour (curved or angular) in two experiments using drawings of common-use objects. The drawings consisted of pairs of the same object with a curved and a sharp-angled version created by quasi-expert students in Design as described in *Bertamini & Sinico (2019)*. They were rated by non-experts for seven characteristics, confirming an association between curvature and beauty. In the current experiments, we examined whether the selection of pairs based on the familiarity of the shape of the objects, and specific individual differences, would modulate preference for contour. Each experiment had two tasks. The first tasks were a liking rating task for the drawings in Experiment 1, and a two-alternative forced-choice (2AFC) task simulating approach/avoidance responses in Experiment 2. The second task was a subjective familiarity selection task for the shape of the objects in both experiments. In this task, participants categorized the object pairs in three groups: (a) the pairs in which the curved shape was the most familiar, (b) the pairs in which the sharp-angled shape was the most familiar, and (c) the pairs in which both shapes were equally familiar. This way, we could analyse preference for curvature in each group. At the end of the experimental tasks, all participants were administered a set of individual measures: a Spanish adapted scale of Art interest and Art

knowledge (*Chatterjee et al., 2010*), the Openness to experience Scale from the NEO-FFI (*McCrae & Costa, 2004*), the items of the Unconventionality facet from the HEXACO personality test (*Lee & Ashton, 2004*), and the Types of Intuition Scale (TIntS) (*Pretz et al., 2014*).

First, we hypothesized that participants would prefer the curved object drawings in both experiments because preference for curvature has shown to be consistent across different stimuli and experimental tasks (*Palumbo & Bertamini, 2016*; *Chuquichambi et al., 2021*). Second, we expected that the curved contours would be perceived as the most familiar because of the predominant role of curvature on shape's perception (*Pasupathy & Connor, 2002*) and its suggested higher exposure in nature (*Koenderink, 1984*; *Hoffman & Singh, 1997*; *Bertamini, Palumbo & Redies, 2019*; *Ruta et al., 2019*). Third, familiarity selection for curved shapes might largely explain preference for curved drawings or only influence this preference. That is, we could find that when the curved shapes are selected as the most familiar, the higher the preference for the curved drawings, or we could find preference for the curved drawings without necessarily perceiving the curved shapes as the most familiar. Fourth, according to the divergences between studies, the variation in people's judgements and stimulus characteristics might explain the inconsistent role of some individual differences in preference for curvature (*Corradi et al., 2019b*). Therefore, the current study aimed to assess to what extent preference for curvature might be explained by familiarity for the shape with which the objects were represented in the drawings and whether this would be modelled by individual differences.

## EXPERIMENT 1

### Materials & methods

#### Participants

Forty-nine adult students (41 female, $M_{age} = 21.3$, $SD_{age} = 4.95$) at the University of the Balearic Islands (UIB) volunteered to participate in the experiment. All participants reported normal or corrected to normal vision and were naïve concerning the experimental hypothesis. They provided written informed consent before the experiment. The experiment was conducted following the code of practice of the APA guidelines, and received ethical approval from the Committee for Ethics in Research (CER) of the UIB (Ref: IB 3828/19 PI).

#### Apparatus and materials

Ninety drawings of familiar objects were selected from the IUAV image database (https://osf.io/cx62j/) (*Bertamini & Sinico, 2019*). The selected stimuli consisted of 45 pairs of drawings. Each pair represented the same object, a curved and a sharp-angled version. These pairs were selected considering that the curved and the sharp-angled versions were similar in terms of size, compression ratio of the file (an index used as a measure of image complexity; *Forsythe, Mulhern & Sawey, 2008*; *Palumbo et al., 2014*), perceived lightness, weight, or style according to the data reported by *Bertamini & Sinico (2019)*. On the other hand, some pairs of drawings differed in how they were made. Thirty pairs were hand-made and 15 were computer-made. Similarly, 15 pairs were shaded and 30 were not shaded.

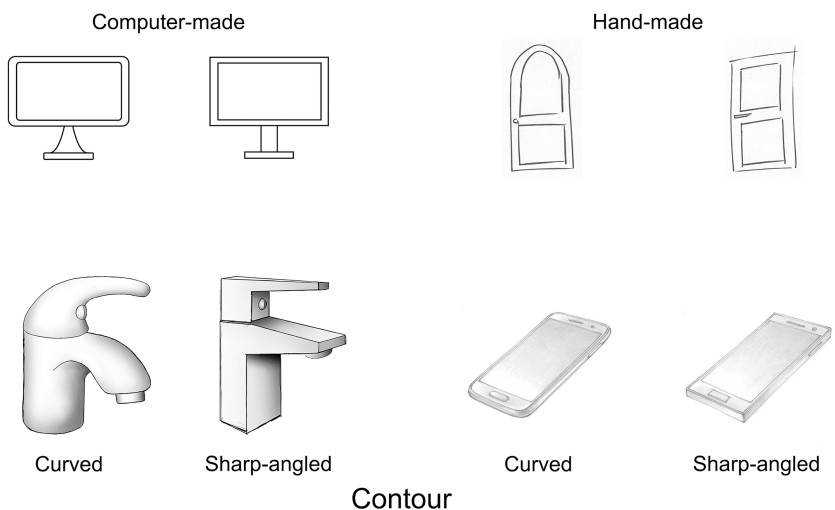

**Figure 1 Examples of the pairs of drawings (IUAV image database).** Each pair has a curved and sharp-angled version. Left-side, computer-made. Right-side, hand-made. Top, not shaded. Bottom, shaded.

Lastly, the apparent position of the objects in relation to the viewer corresponded to a frontal view in 24 pairs, and to a $\frac{3}{4}$ view in 21 pairs. Out of the hand-made drawings, 13 pairs were shaded, 17 pairs were not shaded, 13 pairs were in $\frac{3}{4}$ view, and 17 pairs were in frontal view. Out of the computer-made drawings, 2 pairs were shaded, 13 pairs were not shaded, 8 pairs were in $\frac{3}{4}$ view, and 7 pairs were in frontal view. The curved and the sharp-angled version of each pair had the same Category, Shading and Position. The pairs of stimuli were equalized in size and had 300 dpi resolution. Every stimulus was presented framed on an outline of 600 pixels height, and 600 pixels width. (Fig. 1).

We used the same drawings in the liking rating task and the familiarity selection task. The liking task recorded ratings of each drawing using a horizontal sliding bar from 0 to 100. The ends of the bar had the labels "*I don't like it*" (0) on one side, and "*I like it very much*" (100) on the other side (Fig. 2A). Each stimulus was presented on the centre of the screen until the participant had responded on the sliding bar using the mouse. The task had 8 practice trials corresponding to 4 additional pairs of drawings from the image database, and 90 experimental trials corresponding to the 45 stimuli pairs. Trial sequence was randomized.

The familiarity selection task presented each pair of drawings simultaneously, one on the left and the other on the right side of the screen, until the participant responded. The question was "*Which shape is the most familiar for this object?*" There were three-alternative responses labelled as left, equal, and right. If they chose the shape of the left-side object as the most familiar, they had to press the left key. If they chose the shape of the right-side object, they had to press the right key. They could also choose the shape of both objects as equally familiar by pressing the central key. The task had 8 practice trials and 45 experimental trials corresponding to the 45 pairs. Left-side and right-side presentation and trial sequence were randomized.
A

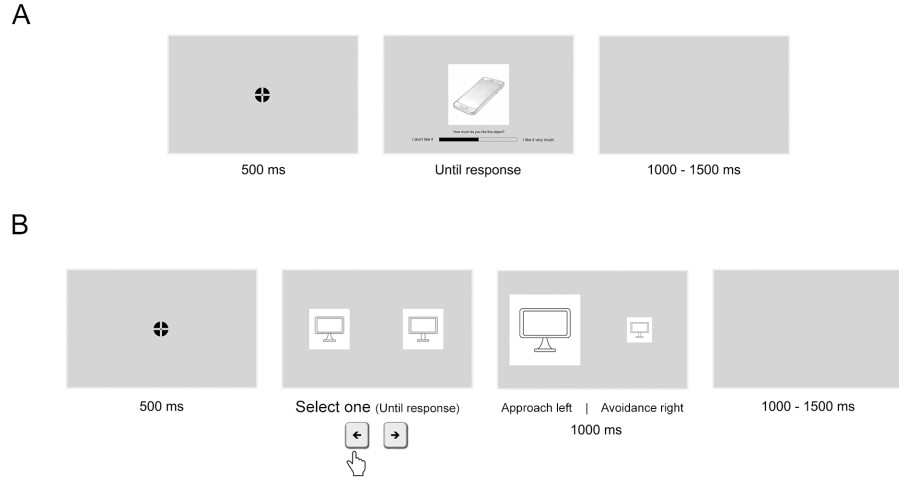

B

**Figure 2 Trials sequence in the preference tasks of experiments 1 and 2.** (A) An example trial in the liking rating task from Experiment 1. (B) An example trial in the two-alternative forced-choice task from Experiment 2. The example shows that the left object was selected. In the next slide, the left object (selected) and the right object (non-selected) simulated approach and avoidance actions, respectively.

Four questionnaires were administered. The first was an Art interest and Art knowledge scale adapted from *Chatterjee et al.*'s (*2010*) Art Training, Interest and Activities Scale. This scale was used in previous studies of aesthetic sensitivity (e.g., *Corradi et al., 2019b*). It consists of eight items with a 0–6 Likert scale. Five items (1–5) measure interest in art, and three (6–8) measure formal education in art. The second questionnaire was the Openness to experience Scale of the NEO-FFI (*McCrae & Costa, 2004*). It consists of twelve items rated on a scale ranging from 1 (*strongly disagree*) to 5 (*strongly agree*). The third questionnaire consists of four items about the Unconventionality facet of the Openness to experience domain from the HEXACO 100 Personality Inventory-Revised (*Lee & Ashton, 2004*). We included this measure because *Cotter et al. (2017)* showed that higher scores on the Unconventionality facet predicted greater preference for curvature using geometrical patterns. Finally, participants completed the Types of Intuition Scale (TIntS) to examine whether the way people make decisions and solve problems modulates preference for drawings (*Pretz et al., 2014*). This scale consists of 23 items (e.g., "I am a 'big picture' person", "I tend to use my heart as a guide for my actions") rated on a scale ranging from 1 (*definitely false*) to 5 (*definitely true*). The items are grouped into four subscales: Holistic Abstract (HA, thinking about a problem in abstract terms), Holistic Big Picture (HB, focusing on the entire problem rather than details of the situation), Inferential (I, making decisions based on automatic, analytic processes), and Affective (A, making decisions by relying on emotional reactions to a situation). The scores of the questionnaires are reported in Table 1.

All tasks were designed with OpenSesame (3.2) software (*Mathôt, Schreij & Theeuwes, 2012*). They were implemented in computers equipped with Intel i5 processors and 21–inch screens set at 1,920 × 1,080 pixels.
**Table 1 Descriptive statistics for the individual differences measures of Experiment 1 ($n = 49$).** Score ranges: Art interest (0–30), Art knowledge (0–18), Openness to experience (11- 60), Unconventionality (4–20), HA (3–15), HB (4–20), I (8–40), A (8–40).

| Variable | Mean | Median | SD | Min –Max |
|---|---|---|---|---|
| Art interest | 10.6 | 12 | 5.44 | 1 –20 |
| Art knowledge | 1.43 | 1 | 2.03 | 0 –11 |
| NEO: Openness to experience | 47.4 | 48 | 5.94 | 30–59 |
| HEXACO: Unconventionality | 3.61 | 3.75 | .57 | 2.25–5 |
| TIntS: Holistic Abstract (HA) | 8.4 | 8 | 2 | 3–14 |
| TIntS: Holistic Big picture (HB) | 13.3 | 13 | 2.47 | 8–19 |
| TIntS: Inferential (I) | 28.5 | 29 | 3.4 | 19–35 |
| TIntS: Affective (A) | 25 | 25 | 5.03 | 16–36 |

### Procedure

The experimental session was carried out at the Psychology Laboratory of the UIB, using isolated cabins and individual computers with the same software and light conditions. Participants were welcomed at the laboratory and they provided written informed consent. They received verbal and written instructions before starting each task. The liking task was the first one. Participants were told that a drawing would be presented at the centre of the computer screen. They had to indicate how much they liked the drawing with a mouse click on the horizontal sliding bar. Next, participants carried out the familiarity selection task. They were told that pairs of drawings would be presented on the computer screen, one on the left and the other on the right side of the screen. They had to select which shape was the most familiar for the object in the drawing, or whether both shapes were equally familiar, by pressing the appropriate key. After these tasks, participants filled in the four questionnaires. The experimental session lasted about 20 min. Finally, participants were debriefed and thanked.

### Data analysis

Data analysis was carried out with the R environment for statistical computing (*R Core Team, 2018*). Participants' responses in the liking task, the familiarity selection task and questionnaires were analysed by means of linear mixed effects models (*Hox, 2010*; *Snijders & Bosker, 2012*). These models account simultaneously for the between-subject and within-subject effects of the independent variables (*Baayen, Davidson & Bates, 2008*). They have been previously used to analyse preference judgements and individual differences (e.g., *Corradi et al., 2019a*; *Corradi et al., 2019b*). The 'lmer' function from the lme4 package was used to fit the models (*Bates et al., 2015*). The afex package (*Singmann et al., 2016*), with the likelihood ratio test, was used to produce the inferential statistics and *p* values. The lsmeans package was used to obtain predicted means for the fixed effects (*Lenth, 2016*). Participant and Stimulus were included as random effects in all models. Model selection was carried out considering model fit indices and following *Barr et al.*'s (*2013*) and *Brauer & Curtin*'s (*2018*) guidelines to choose the maximal random-effects structure justified by the experimental design. Finally, we performed a study of influential cases based on Cook's

distance (Cook's D) in each model. This measure evaluates each participant's influence on the results by examining the impact of its removal from the data set (*Corradi et al., 2018*).

## Results

We considered three models. The first model tested preference for curvature and its relation to the other stimulus properties: computer-made versus hand-made, shaded versus not shaded, and frontal versus $\frac{3}{4}$ view. The second model analysed the relationship between preference for curvature and familiarity selection. The third model tested the influence of the individual measures (i.e., personality and art expertise) on the liking ratings related to preference for curvature.

The first model aimed to predict liking ratings based on Contour (curved vs. sharp-angled), Category (computer-made vs. hand-made), Shading (shaded vs. not shaded), and Position (frontal vs. $\frac{3}{4}$ view) as factors of fixed effects. We also included the interactions between Contour and Category, Contour and Shading, and Contour and Position. The best model, according to models fit indices, included random slopes within participant random effect. Influential cases analysis revealed no influential cases whose value exceeded the recommended cut-off point, which was .090. Participants significantly liked the curved drawings ($M = 55.1$, 95% CI [50–60.2]) more than the sharp-angled ones ($M = 50.4$, 95% CI [45.4–55.4]), $\beta = 3.51$, SE $= 1.5$, t(92.8) $= 2.31$, $p = .023$, 95% CI [.53–6.5] (Fig. 3A). There was no significant interaction of Contour $\times$ Category, $\beta = -1.94$, SE $= 1.52$, t(4217) $= -1.28$, $p = .20$, 95% CI [$-4.9, -1.04$], Contour $\times$ Shading, $\beta = 1.85$, SE $= 1.6$, t(364) $= 1.16$, $p = .24$, 95% CI [$-1.3–5$], or Contour $\times$ Position, $\beta = -2.45$, SE $= 1.4$, t(4217) $= -1.76$, $p = .080$, 95% CI [$-5.2–.30$]. Participants also significantly liked the drawings with shading ($M = 61.7$, 95% CI [54–69.4]) more than the drawings with no shading ($M = 43.8$, 95% CI [38.3–49.3]), $\beta = 17$, SE $= 4.64$, t(64) $= 3.66$, $p < .001$, 95% CI [7.9—-26.1]. There was no significant difference between the hand-made ($M = 50.6$, 95% CI [45.5–55.6]) and the computer-made drawings, ($M = 55$, 95% CI [47.6–62.3]), $\beta = -3.4$, SE $= 4.1$, t(44) $= -.83$, $p = .41$, 95% CI [$-4.6–11.4$]. Similarly, liking ratings did not significantly differ between the drawings in frontal ($M = 50.7$, 95% CI [44.5–57]) and $\frac{3}{4}$ view ($M = 54.8$, 95% CI [48.8–60.7]), $\beta = -2.82$, SE $= 3.7$, t(44) $= -.75$, $p = .45$, 95% CI [$-10.1–4.5$].

The familiarity selection task showed that the curved shapes were selected as the most familiar ones in a proportion of .49, the sharp-angled shapes were selected as the most familiar ones in a proportion of .22, and both shapes were selected as equally familiar in a proportion of .29. The second model included liking rating as the variable to be predicted, and Contour type and Familiarity as categorical fixed effects. The interaction between the two factors was also included, as our main objective was to examine the relationship between contour preference and familiarity. The three familiarity categories were included in the analysis as three levels: the curved shape selected as the most familiar, the sharp-angled shape selected as the most familiar, and both shapes selected as equally familiar. The best model included random slopes within participant and stimulus. Influential cases analysis revealed two influential cases exceeding the recommended cut-off point, which was .087. Therefore, these participants were excluded from the analysis. Results showed that the Contour $\times$ Familiarity interaction was significant when we considered the curved

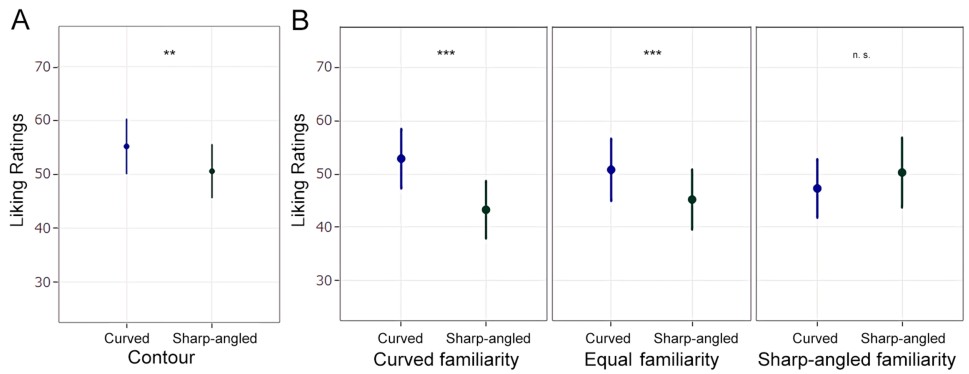

**Figure 3 Liking ratings and familiarity selections of Experiment 1.** (A) Mean liking ratings for the curved and sharp-angled drawings. (B) Mean liking ratings for the drawings within the three alternative responses of the Familiarity selection task. Left graphic represents familiarity selections for the curved shapes, middle graphic represents both shapes selected as equally familiar, and right graphic represents familiarity selections for the sharp-angled shapes. Each one of these graphics show mean liking ratings for the curved and sharp-angled drawings. The curved drawings were liked more when the curved shapes were selected as the most familiar ones, or when both shapes were selected as equally familiar, but not when the sharp-angled shapes were selected as the most familiar ones. Error bars represent 95% CI ( ** $p \leq .01$ , *** $p \leq .001$, n.s.: not significant).

and sharp-angled responses in the Familiarity factor, $\beta = 12.58$, SE $= 2.22$, t(82.5) $=$ 5.67, $p < .001$, 95% CI [8.23–16.93]. Specifically, participants liked the curved drawings ($M = 52.9$, 95% CI [47.3–58.4]) more than the sharp-angled ones ($M = 43.3$, 95% CI [38, 48.7]) when the curved shapes were selected as the most familiar ones, $\beta = 9.6$, SE $= 1.6$, t(44) $= 5.8$, $p < .001$. Conversely, when the sharp-angled shapes were selected as the most familiar ones, liking ratings did not differ significantly between the sharp-angled drawings ($M = 50.3$, 95% CI [43.7–56.8]) and the curved ones ($M = 47.3$, 95% CI [41.8–52.8]), $\beta = 3$, SE $= 2.1$, t(23.7) $= 1.4$, $p = .16$ (Fig. 3B). The Contour ×Familiarity interaction was significant when we considered both shapes selected as equally familiar and the sharp-angled shapes selected as the most familiar ones, $\beta = 8.46$, SE $= 2.2$, t(79) $= 3.79$, $p < .001$, 95% CI [4.1–12.8]. This effect revealed that when both the curved and sharp-angled shapes were selected as equally familiar, participants still liked the curved drawings ($M = 50.8$, 95% CI [45, 56.6]) more than the sharp-angled ones ($M = 45.2$, 95% CI [39.5–50.8]), $\beta = 5.6$, SE $= 1.5$, t(25) $= 3.6$, $p = .0010$. Lastly, the Contour x Familiarity interaction also reached significance when we considered the curved shapes selected as the most familiar ones and both shapes selected as equally familiar, $\beta = 4.01$, SE $= 1.87$, t(102.1) $= 2.15$, $p = .034$, 95% CI [.36–7.67]. In conclusion, we found an effect of preference for curvature when the curved shapes were selected as the most familiar ones and when both shapes were selected as equally familiar. However, there was no effect of preference for contour when the sharp-angled shapes were selected as the most familiar ones.

Regarding the individual measures, we analysed whether they modulated liking ratings related to the curved and sharp-angled drawings. The model predicted liking ratings based on Contour and its interactions with Art interest, Art knowledge, Openness to experience, the Unconventionality facet, and TIntS subscales (HA, HB, I and A) as predictors. All

continuous predictors were centred on the grand mean. The best model included random slopes within participant and stimulus. Influential cases analysis showed no influential cases whose value exceeded the recommended cut-off point, which was .10. Results revealed that participants who scored higher in the Holistic Big Picture Subscale (HB) showed higher liking ratings for all the drawings, $\beta = 1.3$, SE $= .52$, t(23) $= 2.5$, $p = .020$, 95% CI [.28–2.32]. All other effects and interactions were nonsignificant. All effects are included in Table S1 as supplementary material.

## Discussion

Experiment 1 showed that participants liked the curved drawings more than the sharp-angled ones. This result supports the curvature effect (*Corradi & Munar, 2020*). Our results also reported an interaction between familiarity and curvature on shape preference. When the curved shapes were selected as the most familiar ones, the curved drawings were liked more than the sharp-angled drawings. This finding supports the role of familiarity in predicting aesthetic preference (*Verhaeghen, 2018*; *Chmiel & Schubert, 2019*). That is, the drawings with the shapes that were chosen as most familiar to represent the objects were liked more. However, we also found that when the shapes of the objects were selected as equally familiar, participants also liked the curved drawings more than the sharp-angled ones. Furthermore, when the sharp-angled shapes were selected as the most familiar ones, liking did not differ between the curved and sharp-angled drawings. Altogether, these findings suggest that familiarity of the shape with which the objects have been represented in the drawings modulates preference for curvature, but it does not completely explain participants' preference for the curved drawings.

Individual measures analysis showed that participants with higher scores in the Holistic Big Picture subscale liked all the drawings more than participants with lower scores. All the other measures did not significantly influence liking ratings. These findings are in line with studies suggesting an uncertain role of some individual measures on preference for curvature (*Corradi et al., 2019b*).

## EXPERIMENT 2

Experiment 2 consisted of a 2AFC task simulating approach/avoidance responses (Fig. 2B). Approach/avoidance procedures have been previously used in preference for curvature research (*Vartanian et al., 2013*; *Palumbo, Ruta & Bertamini, 2015*). Participants carried out the same familiarity selection task and questionnaires as in Experiment 1. In the 2AFC task, each pair of drawings was presented on the screen until participants responded, as in previous studies (*Munar et al., 2015*; *Corradi et al., 2018*). However, although these studies reported preference for images of curved real objects in short and medium presentation times, the effect disappeared in the until-response condition. Similarly, these authors reported preference for curved abstract patterns in short and medium presentation times, but in this case, the effect increased in the until-response condition. *Palumbo & Bertamini (2016)* showed that preference for curvature was consistent across tasks using irregular shapes. Considering these studies and the results from Experiment 1, we expected that participants would prefer the curved object drawings more than the sharp-angled ones.

**Table 2  Descriptive statistics for the individual differences measures of Experiment 2 ($n = 49$).** Score ranges: Art interest (0–30), Art knowledge (0–18), Openness to experience (12 60), Unconventionality (4 20), HA (3–15), HB (4–20), I (8–40), A (8–40).

| Variable | Mean | Median | SD | Min –Max |
|---|---|---|---|---|
| Art interest | 10 | 9 | 6.22 | 0–26 |
| Art knowledge | 2.35 | 1 | 2.94 | 0–12 |
| NEO: Openness to experience | 46.3 | 45 | 5.53 | 36–58 |
| HEXACO: Unconventionality | 3.58 | 3.5 | .58 | 2.5–5 |
| TIntS: Holistic Abstract (HA) | 7.94 | 8 | 2.21 | 3–13 |
| TIntS: Holistic Big picture (HB) | 12.8 | 12 | 2.55 | 7–20 |
| TIntS: Inferential (I) | 29.5 | 30 | 3.33 | 20–38 |
| TIntS: Affective (A) | 25.3 | 25 | 5.53 | 12–35 |

Furthermore, we expected that shape familiarity would also modulate preference for curvature.

## Materials & Methods
### Participants
Forty-nine adult students (35 female, $M_{age} = 26.3$, $SD_{age} = 6.5$) at the UIB volunteered to participate in the experiment. All participants reported normal or corrected to normal vision and were naïve concerning the experimental hypothesis. They provided written informed consent before the experiment and were treated following the code of practice of the APA guidelines. The study received ethical approval from the Committee for Ethics in Research (CER) of the UIB (Ref: IB 3828/19 PI).

### Apparatus and materials
We used the same 90 drawings as in Experiment 1 (Fig. 1). They were presented both in the 2AFC task and the familiarity selection task. In the 2AFC task, each pair of stimuli was presented until response, a drawing on the left and the other on the right side of the computer screen (Fig. 2B). Participants were instructed to select one of the two object drawings, and instructions avoided the words 'liking', 'wanting' and 'preference' as in *Munar et al. (2015)* and *Corradi et al. (2018)*. Later, the selected drawing was enlarged to twice its previous size, while the non-selected one was shrunk to half its previous size at the same position for 1,000 ms. This action simulated an approach/avoidance behaviour (*Bamford et al., 2015*). As in Experiment 1, the 2AFC task had 8 practice trials with additional stimuli from the image database, and 45 experimental trials corresponding to the 45 pairs of drawings. Left-side and right-side stimulus presentation and trial sequence were randomized. The familiarity selection task and the set of questionnaires were the same as in Experiment 1. The scores of the questionnaires are reported in Table 2.

### Procedure
The experimental session was carried out as in Experiment 1. Participants received verbal and written instructions before starting each task. First, they carried out the 2AFC task. They were told that they had to select one of two drawings presented on the screen using the right and left arrow keys. Then, the size of the selected drawing would be enlarged, and

the size of the non-selected drawing would be shrunk. Next, they carried out the familiarity selection task receiving the same instruction as in Experiment 1. Lastly, they filled in the questionnaires using the same computer. The experimental session lasted about 20 min. Finally, participants were debriefed and thanked.

### Data analysis

Analyses were carried out with the R environment for statistical computing (*R Core Team, 2018*). We mainly modelled responses by means of generalized linear mixed effects models given that the dependent variable in the 2AFC task was the kind of contour participants selected (curved or sharp-angled). The 'glmer' function from the lme4 package was used to fit the models (*Bates et al., 2015*). All models included Participant and Stimulus as random effects. Model selection was performed following the same considerations outlined in Experiment 1. Finally, we performed a study of influential cases in each model.

## Results

We considered three analyses. First, we analysed preference for curvature and its relationship with the other stimulus characteristics. The second analysis was based on a model to test the relationship between preference for curvature and familiarity selection. The third analysis examined the influence of the individual measures on preference for curvature.

Previously, we carried out a *t*-test on the preference for curvature as compared to angularity to examine participants preference choices in the 2AFC. Results showed that participants chose the curved drawings significantly above chance level (M = .61), $t$ (48) = 5.54, $p < .001$, 95% CI [.57–.65], $d = .79$ (Fig. 4A). Next, we modelled the curved choices as the variable to be predicted. The model included Category (computer-made vs. hand-made), Shading (shaded vs. not shaded), Position (frontal vs. $\frac{3}{4}$ view), and the interaction between these factors as fixed effects. The best model included random intercepts within participant and stimulus. Influential cases analysis revealed no influential values exceeding the recommended cut-off point, which was .089. Results revealed no significant effect either for Category, $\beta = -1.38$, SE = .76, Z = $-1.81$, $p = .070$, 95% CI [$-2.9$–.11], Shading, $\beta = .02$, SE = .53, Z = .04, $p = .96$, 95% CI [$-1.02$–1.06], or Position, $\beta = -.74$, SE = .53, Z = $-1.4$, $p = .16$, 95% CI [$-1.8$–.30]. Moreover, there was no significant interaction between Category × Shading, $\beta = .27$, SE = .94, Z = .29, $p = .77$, 95% CI [$-1.57$–2.1], Category ×Position, $\beta = .90$, SE = .71, Z = 1.26, $p = .21$, 95% CI [$-.50$, 2.3], or Shading x Position, $\beta = .75$, SE = .72, Z = 1.04, $p = .30$, 95% CI [$-.66$, 2.15]. These results indicated that the choice of the curved drawing does not depend on the category of the drawing, whether or not it is shaded, and whether it is in frontal or $\frac{3}{4}$ view.

On the other hand, the familiarity selection task showed that the curved shapes were selected as the most familiar in a proportion of .45, the sharp-angled shapes were selected as the most familiar in a proportion of .21, and both shapes were selected as equally familiar in a proportion of .34. We modelled whether familiarity selection predicted preference in the 2AFC task. The model included curved choices as the variable to be predicted.
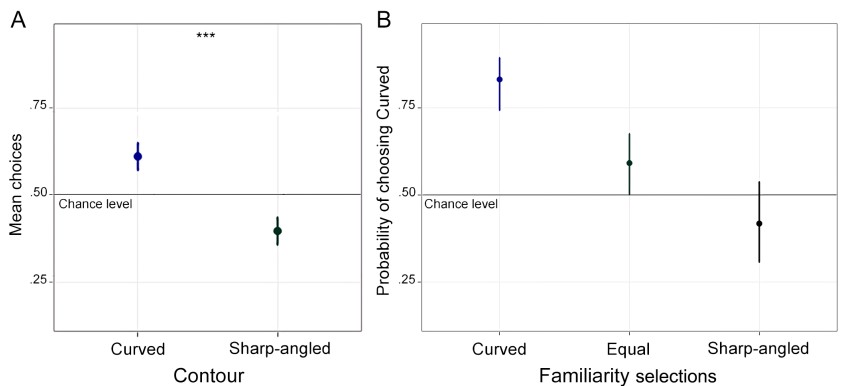

**Figure 4** **Preference choices and familiarity selections of Experiment 2.** (A) Mean choices of the curved and sharp-angled drawings in the 2AFC task. (B) Probability of choosing the curved drawings in the 2AFC task within the three alternative responses of the Familiarity selection task. Familiarity selections for the curved shapes and both shapes selected as equally familiar predicted a higher probability of choosing the curved drawings in the 2AFC. Error bars represent 95% CI ( ***$p < .001$).

Familiarity (curved, equally, sharp-angled) and Lateralization (left vs. right) were included as categorical fixed effects. The best model included random slopes within participant. Influential cases analysis revealed no extreme values exceeding the recommended cut-off point, which was .087. Results showed a main effect when we compared the pairs in which the curved shape was the most familiar and the pairs in which the sharp-angled shape was the most familiar, $\beta = 1.90$, SE $= .35$, $Z = 5.5$, $p < .001$, 95% CI [1.22–2.6]. Post-hoc tests revealed that curved preference was higher when the curved shapes were selected as the most familiar (M $= .83$, 95% CI [.74–.89]) than when the sharp-angled shapes were selected as the most familiar (M $= .42$, 95% CI [.31–.53]), OR (Odds Ratio) $= 6.72$, 95% CI [4.1–13.2]. That is, when participants selected the curved shapes as the most familiar ones, they also mostly preferred the curved drawings over the sharp-angled ones in the 2AFC task, but this was not the case when participants selected the sharp-angled shapes as the most familiar ones. Similarly, there was a main effect when we considered the curved shapes selected as the most familiar ones and both shapes selected as equally familiar, $\beta = 1.22$, SE $= .25$, $Z = 4.96$, $p < .001$, 95% CI [.74–1.71]. Curved preference choices were higher when the curved shapes were selected as the most familiar ones than when both shapes were selected as equally familiar (M $= .59$, 95% CI [.50–.67]), OR $= 3.4$, 95% CI [2.1–5.5]. Lastly, there was a main effect when we considered both shapes selected as equally familiar and the sharp-angled shapes selected as the most familiar ones, $\beta = .68$, SE $= .23$, $Z = 2.9$, $p = .0035$, 95% CI [.22–1.14]. Post-hoc comparisons showed that curved preference was higher when participants selected both shapes as equally familiar than when they selected the sharp-angled shapes as the most familiar ones, OR $= 1.98$, 95% CI [1.25–3.12] (Fig. 4B). These results suggest that participants preferred the drawings they chose as more familiar. They also support the findings from Experiment 1, suggesting that shape familiarity modulates preference for curvature between tasks.

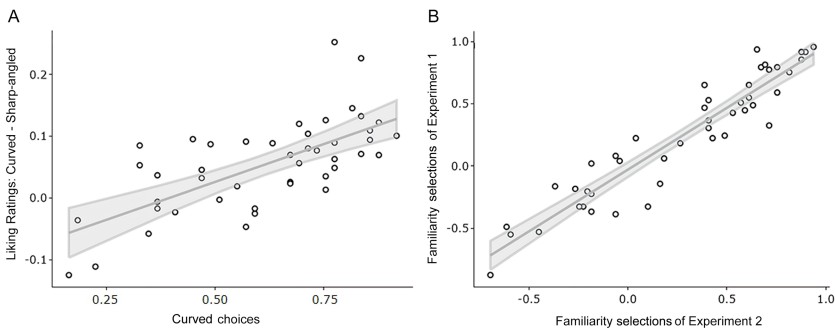

**Figure 5** **Scatterplots showing the relation between the data from experiments 1 and 2.** (A) Relation between the liking ratings (Experiment 1) and the curved choices in the 2AFC (Experiment 2). (B) Relation between the familiarity selections data of Experiment 1 and 2. Each point represents a pair of drawings. All $p$'s < .001.

Regarding the individual measures, we modelled whether they modulated contour preference choices. The model included Art interest, Art knowledge, Openness to experience, Unconventionality facet, and TIntS subscales (HA, HB, I and A) as predictors. These predictors were centred on the grand mean. The best model included random slopes within participant and stimulus. Influential cases analysis revealed four influential values exceeding the recommended cut-off point, which was .10. Thus, these participants were excluded from the analysis. Results showed that participants who scored higher in the HB subscale showed a significantly higher preference for curved drawings, $\beta = .13$, SE = .04, $Z = 3.5$, $p < .001$, 95% CI [.06–.20]. Those who scored higher in the A subscale also showed a significantly higher preference for curved drawings, $\beta = .052$, SE = .016, $Z = 3.3$, $p < .001$, 95% CI [.02–.08]. In contrast, participants who scored higher in the Unconventionality facet showed significant lower preference for curved drawings, $\beta = -.09$, SE = .04, $Z = -2.2$, $p = .028$, 95% CI [$-.17$ to $-.01$]. The other effects were nonsignificant. All effects are included in *Table S2* as supplementary material.

### *Correlations between experiments*

We analysed the correlation between the data from the two experiments to determine the consistency of responses to the same drawings from different participants. First, we performed a correlation analysis based on drawings between liking ratings in Experiment 1 and preference choices in Experiment 2. From Experiment 1, we calculated the difference between the liking for the curved drawing and the sharp-angled drawing of each pair. We correlated these values with the choice mean (between 0 a 1) for each pair of drawings from the 2AFC task in Experiment 2. Subsequently, the bias-corrected and accelerated CI was calculated using the bootstrap resampling method with 1499 samples suggested for a test at the 0.05 and 0.01 level (*Davidson & MacKinnon, 2000*). Results revealed a significant positive correlation between the liking ratings and curved preference choices, $r_s(45) = .66$, $p < .001$, 95% CI [.46–.80]. This result supported a positive relationship of preference for drawings between tasks (Fig. 5A).

Second, we compared the familiarity responses of participants in the two experiments. We obtained a familiarity value for each pair of stimuli regarding the three-alternative responses from the familiarity selection tasks. That is, we grouped the trials where participants selected the curved shape as the most familiar (+1) sharp-angled shape as the most familiar (−1) and both shapes as equally familiar (0) to obtain a familiarity value between −1 and 1 for each pair of stimuli. Then, we correlated these values between both familiarity selection tasks and calculated CI as in the first correlation. Results showed a strong positive association of familiarity judgements between the two experiments, $r_s(45)$ = .92, $p < .001$, 95% CI [.88, .94]. These results supported that familiarity with the shape of the objects was consistent across different participants (Fig. 5B).

### Discussion

Experiment 2 showed that participants preferred the curved drawings over the sharp-angled ones. This result supported our main hypothesis about the curvature effect (*Corradi & Munar, 2020*). Therefore, together with the results from Experiment 1, we suggest a consistent preference for the curved drawings of common-use objects between tasks.

We also found that familiarity for the curved stimuli predicted a higher preference for curvature in the 2AFC task than familiarity for the sharp-angled stimuli and the stimuli selected as equally familiar. That is, when the curved shapes were selected as the most familiar ones, there was a higher preference for curvature. Similarly, there was preference for curvature when participants selected both shapes as equally familiar. In contrast, we did not find preference for angularity or for curvature when participants selected the sharp-angled shapes as the most familiar ones. These results support the influence of familiarity on preference. However, they also showed that familiarity is not the only factor determining preference for drawings of common-use objects.

On the other hand, some individual measures influenced preference choices. Specifically, participants who scored higher in the HB and A subscales of the TIntS showed a higher preference for the curved drawings. In contrast, those who scored higher in the Unconventionality facet showed less preference for the curved drawings. These results suggest that the 2AFC task is a more sensitive procedure to find the potential influence of individual differences in preference for curvature than the liking rating task. Conversely, the results also suggest an uncertain influence of some individual measures (e.g., art expertise or openness to experience) on preference for curvature (*Corradi et al., 2019b*).

Finally, the correlation analysis between the data from the two experiments showed a similar pattern of preference for the pair of drawings. On the other hand, the perception of familiarity with the shape of the objects and their representational content was highly consistent using two different groups of participants.

## GENERAL DISCUSSION

We examined preference for curvature and its relationship with familiarity using drawings of common-use objects in two experiments. Experiment 1 consisted of a liking rating task, a familiarity selection task, and a set of individual measures. Experiment 2 used the same stimuli and different participants, and consisted of a 2AFC task simulating

approach/avoidance responses, and the same familiarity selection task and individual measures of Experiment 1.

In Experiment 1, we found higher liking ratings for the curved than the sharp-angled drawings. Similarly, in Experiment 2, participants preferred the curved drawings over chance level in the 2AFC task. These findings support the curvature effect using drawings of common-use objects (*Corradi & Munar, 2020*). They also support the preference for curvature as a consistent effect between different experimental designs (*Palumbo & Bertamini, 2016*; *Chuquichambi et al., 2021*). Conversely, our findings diverge from those of some previous studies using images of real-objects. *Munar et al. (2015)* did not find preference for curved objects in a 2AFC task in the until-response condition. Similarly, using the same task and stimuli than *Munar et al. (2015)*, *Corradi et al. (2018)* found that the effect of preference for curvature decreased as the presentation time increased. They suggested a higher influence of the meaning and content-related information of stimuli as the presentation time increased. In this regard, they found that the effect of preference for curvature was stronger when presenting abstract patterns in longer presentation time compared to brief presentations. With Japanese participants, *Maezawa, Tanda & Kawahara (2020)* did not find a preference for curvature using similar stimuli as *Corradi et al. (2018)* and like/dislike and rating scale tasks. A possible explanation of these divergences may be related to the interaction between the meaningful and representational content of the object, familiarity with its shape, and the artistic view of the drawings because of their design and artistic nature (*Schroll, Schnurr & Grewal, 2018*).

The curved drawings were mostly preferred when the curved shape was selected as the most familiar or when the two shapes were selected as equally familiar, but not when the sharp-angled shape was selected as the most familiar. Further, in both experiments, preference for the curved drawings was higher when the curved shape was selected as the most familiar than when both shapes were selected as equally familiar. These findings support familiarity as a strong predictor of preference (*Reber, Winkielman & Schwarz, 1998*; *Reber, Schwarz & Winkielman, 2004*; *Verhaeghen, 2018*; *Chmiel & Schubert, 2019*). However, they also suggest that familiarity is not the only factor determining preference for curvature because participants still preferred the curved drawings over the sharp-angled ones when the two shapes of the objects were selected as equally familiar. Moreover, there was no preference for the sharp-angled drawings when the sharp-angled shape was selected as the most familiar. In addition, these findings might support curvature as one of the diverse fluency-enhancing variables that explains the relationship between prototypicality and attractiveness (*Winkielman et al., 2006*).

Our results on the relationship between preference for curvature and familiarity might also be connected to the predominant role of curvature on shape's perception (*Pasupathy & Connor, 2002*). Our visual system might integrate curved features more efficiently because they tend to match the statistic regularities of the natural environment (*Sigman et al., 2001*; *Bertamini, Palumbo & Redies, 2019*; *Stanischewski et al., 2020*). Relatedly, our results might also be explained because of a higher frequency of curved contours in natural scenes (*Ruta et al., 2019*). In a recent study, *Yue, Robert & Ungerleider (2020)* found a specialized cortical network for curvature processing in humans. They suggested the

interaction between preference for curvilinearity with central-peripheral processing biases as an important organizing principle for temporal cortex topography. Interestingly, they also found a possible link between curvature-preferring areas and face-selective areas. This study also dealt with curvature as a metric property. However, *Amir, Biederman & Hayworth (2011)*; *Amir, Biederman & Hayworth (2012)* showed greater sensitivity to the non-accidental property related to the difference between curved and straight contours than to the metric property of curvature. Our brain might represent the non-accidental property in a different way than the metric property of curvature. Altogether, these studies and the interaction between the representational nature of the objects and the artistic characteristics of the drawings within the same stimuli may explain our results of the role of familiarity in preference for curvature.

However, the current research line on preference for curvature leaves open the role that could play the phenomenology of how space appears to the perceiver. In particular, some artists use a curvilinear perspective, instead of a linear perspective because it is closer to the viewer's experience that straight lines in nature can be perceived as curved ones(*Pepperell, 2012*; *Pepperell & Haertel, 2014*). On the other hand, a curved line can even appear as a straight line when viewed head-on, or circles in the peripheral visual field can appear polygonal in shape (*Baldwin et al., 2016*). Further research is needed to address this issue.

Besides the role of object characteristics, previous studies reported that individual measures also can modulate preference for curvature (e.g., *Cotter et al., 2017*; *Silvia & Barona, 2009*). In Experiment 1, we only found that higher scores in the HB subscale predicted higher preference for all the drawings. However, we found some individual differences in Experiment 2, which leads us to suggest that the 2AFC task is more sensitive to finding them than the liking rating task. Specifically, participants with higher scores in the HB and A subscales showed a higher preference for curvature. The influence of the HB type of intuition in preference for curvature might be explained because curved contours facilitate fluent global processing of the stimuli (*Reber, Schwarz & Winkielman, 2004*; *Gómez-Puerto, Munar & Nadal, 2016*). On the other hand, the relationship between the A subscale and the preference for curvature could result from associations with positive valence underlying preference for curvature (*Palumbo, Ruta & Bertamini, 2015*).

Our results also showed that higher scores in the Unconventionality facet predicted less preference for the curved drawings in Experiment 2. This might be related to the idea that the sharp-angled shapes are perceived as more avant-garde (*Ruta et al., 2021*) and unconventional people tend to show a higher preference for innovative designs. Interestingly, *Cotter et al. (2017)* found that higher unconventionality scores predicted more preference for curvature using irregular polygons. However, they found no effect using arrays of circles and hexagons. Using the same arrays of circles and hexagons, *Silvia & Barona (2009)* found preference for curvature in participants without art training –probably more conventional people–but there was no effect with art-trained participants –probably more unconventional people. Artists may show more unconventional thinking and express it in their art because this may make their work more impactful than more conventional artistic styles (*Stamkou, Van Kleef & Homan, 2018*). Conversely, these authors found preference for curvature in art-trained participants but not in participants without

training when they rated complex polygons. Considering these studies, preference for curvature might be higher in art-trained and unconventional participants when the stimuli are more complex. However, we found no influence on preference for curvature from the Art interest and Art knowledge Scales, as in *Corradi et al. (2019b)*. These authors reported that the influence of art interest and art knowledge on aesthetic sensitivity was inconsistent. Altogether, our findings suggest that the influence of individual differences in preference for curvature might depend on the kind of stimuli.

On the other hand, we found significant positive correlations between the results of both experiments. The difference in liking ratings between curved and sharp-angled drawings (Experiment 1), and the preference choices for the curved drawings (Experiment 2) showed a similar pattern of preference. This finding supports a consistent and predictable preference for drawings as representational images (*Vessel & Rubin, 2010*; *Schepman, Rodway & Pullen, 2015*; *Schepman, Rodway & Pullen, 2015*). Although drawings also have art-related characteristics, our results indicate that these characteristics did not weaken the preference consistency between participants. On the other hand, the highly positive correlation between the familiarity selection tasks endorse a robust concept of familiarity of object drawings regardless of the participants.

A possible limitation of this study is that we used a subjective measure of familiarity. The familiarity values came from the direct response of the shape participants considered familiar for the objects. Previous studies used measures based on the exposure time or the number of presentations of the stimulus, that is, a process of familiarization (e.g., *Berlyne, 1970*; *Berlyne, 1971*; *Tinio & Leder, 2009*). However, *Sluckin, Hargreaves & Colman (1982)* argued that subjective measures of familiarity, compared to objective measures, might be more suitable because of a larger variance within each individual and stimulus. Moreover, the drawings involved content-related information. Repeated exposure would likely lead to habituation and, as a consequence, preference could decline (*Biederman & Vessel, 2006*). Using subjective measures, participants only need a single presentation of the stimulus to evaluate its representational content as more or less familiar. Moreover, the subjective familiarity of the shape of an object might be modulated by participant's individual differences. Thus, future studies could assess the role of individual differences on shape familiarity in order to complement our findings of the relationship between these variables with preference for curvature.

## CONCLUSIONS

In summary, we found preference for curvature using drawings of common-use objects in two experiments. The curved shapes of the objects were also selected as the most familiar ones in both experiments. When the curved shapes were selected as the most familiar, and when both shapes were selected as equally familiar, participants showed preference for the curved drawings. However, when the sharp-angled shapes were selected as the most familiar, participants did not show preference for curvature or for angularity. These findings support the idea that shape familiarity modulates preference for drawings of common-use objects. However, they also indicate that the influence of familiarity is not the only factor

explaining the preference for curved drawings. The influence of individual differences in preference for the drawings suggested that the kind of stimuli and the experimental task may predict divergencies across studies and measures. Correlation analyses between experiments also supported a consistent relationship of preference between tasks, and a coherent concept of familiarity of the same pair of object drawings. Altogether, our findings endorse the curvature effect using drawings of common-use objects and familiarity as an important predictor of preference.

## ACKNOWLEDGEMENTS

We are grateful to Marco Bertamini, and Michele Sinico for making the stimuli set available.

### Funding

This research was supported by grant PSI2016-77327-P, awarded by the Spanish Ministerio de Ciencia, Innovación y Universidades, the Agencia Estatal de Investigación (AEI) and the European Regional Development Funds (ERDF). Erick G. Chuquichambi's work was supported by the predoctoral contract FPU18/00365, awarded by the Ministerio de Ciencia, Innovación y Universidades. The funders had no role in study design, data collection and analysis, decision to publish, or preparation of the manuscript.

### Grant Disclosures

The following grant information was disclosed by the authors:
Ministerio de Ciencia, Innovación y Universidades, Agencia Estatal de Investigación (AEI) and European Regional Development Funds (ERDF): PSI2016-77327-P.
Ministerio de Ciencia, Innovación y Universidades: FPU18/00365.

### Competing Interests

The authors declare there are no competing interests.

### Author Contributions

- Erick G. Chuquichambi conceived and designed the experiments, performed the experiments, analyzed the data, prepared figures and/or tables, authored or reviewed drafts of the paper, and approved the final draft.
- Letizia Palumbo and Carlos Rey conceived and designed the experiments, analyzed the data, prepared figures and/or tables, authored or reviewed drafts of the paper, and approved the final draft.
- Enric Munar conceived and designed the experiments, analyzed the data, prepared figures and/or tables, authored or reviewed drafts of the paper, and approved the final draft.

## Human Ethics

The following information was supplied relating to ethical approvals (i.e., approving body and any reference numbers):

The study received ethical approval from the Committee for Ethics in Research (CER) of the UIB (Ref: IB 3828/19 PI).

## Data Availability

The datasets generated in the experiments and the dataset used for correlation analyses are available in the Supplemental Files.

The raw code and raw datasets are available on the Open Science Framework: Chuquichambi, Erick G, Letizia Palumbo, Enric Munar, and Carlos Rey. 2021. ''Shape Familiarity Modulates Preference for Curvature in Drawings of Common-Use Objects.'' OSF. DOI: 10.17605/OSF.IO/XGJ690.

## Supplemental Information

Supplemental information for this article can be found online at http://dx.doi.org/10.7717/peerj.11772#supplemental-information.

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
