# Peer review of "Shape familiarity modulates preference for curvature in drawings of common-use objects"

_PeerJ, doi:10.7717/peerj.11772_

## Round 0.1 · original submission · Major Revisions

Thanks for your submission. I have read through the manuscript myself and obtained reviews from 3 reviewers (R1, R2 and R3). I'd like to ask that you respond to the points raised by the reviewers, particularly the following:

- R1 concerns about how the drawings are described, particularly three-dimensionality and contrast

- R3 concerns about model specification and selection. I agree with the reviewer that aspects of the models used are unclear.

- Please also note that R1 has included an annotated version of the manuscript where they have put additional questions.

In addition, I have a few comments myself:

- I would like to ask that you add some discussion about the relative effect sizes that you observed. I think it is notable that the effects of familiarity were generally much larger than those observed for curved vs. straight.

- I have a thought that may be relevant to your arguments about the familiarity of curved vs. straight lines and their probability of occurrence in nature. It may be true that there are more curved than straight lines (though we live in a very indoor, linear environment). However, a straight line always remains straight, no matter what angle it is viewed from. A curved line, however, is a less reliable signal: it changes its degree of curvature when viewed from a different angle, and can even appear as straight when viewed head-on. It is thus likely that the brain represents this non-accidental difference between straight vs. curved in one manner, but may represent _degree_ of curvature, a metric property, quite differently. Perhaps this has little bearing on your conclusions, but I thought you may find it worth considering.

a few other comments:
ln 36: what is a 'hedonic' product?
ln 155-158: sentence is awkward
ln. 298-300: wording of this result is difficult to follow
ln. 418-419: again hard to follow

·

Basic reporting

All standards listed here are met.
Some minor points in formulation are marked in the attached PDF.
Please provide an accurate definition of what you mean by "drawings".

Experimental design

All standards listed here are met.
See some minor points in the attached PDF.
Please describe whay you did not evaluate the familarity of objects depicted.
Please describe whay the aspect of three dimensionality was not assessed as an image feature in the drawings.

Validity of the findings

All standards listed here are met.
Please provide effect sizes for the results where appropriate.

Additional comments

The manuscript is well written and provides an insight into the research field.
The research was carried out and described accurately.

The definition of the drawings and the categorizations in hand-made, computer-made, shaded, not shaded is of course not exhaustive. Relevant seems to me the aspect of three-dimensionality, which was not or too little considered. Aspects of contrast in the drawing, which also has an effect on liking, could also have been taken into consideration.

Reviewer 2 ·

Basic reporting

no comment

Experimental design

no comment

Validity of the findings

I think the direction of reasoning is opposite, because participants did the preference (or 2AFC) task first followed by the familiarity task. Therefore, participants’ familiarity judgment might be contingent on the preference (or 2AFC selection) made before the familiarity judgements. Therefore, the authors cannot conclude that the familiarity judgements modulate preference for curvature between tasks. Instead, preference judgments could modulate familiarity for objects. If the order of testing does not matter, then the same pattern of the results should be obtained when running the experimental tasks in the reversed order.

Additional comments

Why the authors collected 49 participants in the two experiments?

The authors note that the participants of the present study were treated in compliance with Declaration of Helsinki. If so, the descriptions related to the preregistration should be reported, because research registration is required in a publicly accessible database before recruitment of the first subject (#35).

Reviewer 3 ·

Basic reporting

Mostly sound and clear, background well defined, references well chosen. See 'comments for the authors' for details.

Experimental design

Relevant, clear and sound. See 'comments for the authors' for details.

Validity of the findings

The data are provided, clear and consistent. However, the statistical analysis is unclear and potentially erroneous. This is my main concern. Please see 'comments for the authors' for details and suggestions for improvement.

Additional comments

In this study, the authors explored the interaction between familiarity and preference for curvature. The rationale of the authors is as follows. Preference for curvature has been consistently found in a wide range of experiments using different types of stimuli. However, curved objects are prominent in nature. We are therefore more familiar with curved than sharp-angled objects. As we tend to prefer visual features we are more familiar with, it is therefore possible that preference for curvature is not driven by curvature in itself, as a purely geometrical characteristic, but by the familiarity conveyed by curved objects. Hence, to understand better what drives preference for curvature, we need to spell out the interaction between familiarity and curvature. The research question raised by the manuscript is well delineated, relevant and meaningful.

While I find the experimental paradigm chosen by the authors to assess the interaction between familiarity and preference for curvature sound, I have a major concern, which I think should be tackled before publication.

Statistics -
My main concern is about the statistical analysis. Using mixed effects models is well-suited for analysing the type of data provided by the current experiment, which is a good point for the manuscript. However, I find the description of the mixed modelling used in both experiments incomplete, unclear, and, unless I am misunderstood, incorrect in Experiment 2. This obviously is an important concern as all the conclusions of the study are driven by the statistical analysis. To reassure the authors, I would like to say that solving this point may only be a matter of making the statistical inference explicit and correcting the model in Experiment 2. All the choices for the statistical models are justified stating ‘The maximal model that converged included participant and stimulus as random effects, as well as random slopes within participant’ (l. 266 and throughout). I am not sure I follow the procedure for model selection. The strength of mixed effects models is to allow one to draw general conclusions by controlling the idiosyncrasies of individual observers or stimuli. In the study, ‘participant’ and ‘stimulus’ should be considered as random effects whatever the choice for all the other sides of the modelling, or potential convergence. Choosing whether the intercept, slope, or both should be random can indeed be exploratory, but not whether ‘participant’ and ‘stimulus’ are random effects. What kind of convergence do the authors refer to for choosing the model? Maximal in which sense? A standard procedure for choosing a model is to compare different models in terms of information theoretic criteria (AIC, BIC or others). If the authors have followed a less conventional but sound method they should make it explicit in the manuscript. A plus would be to publish the code used to carry the statistical analysis alongside the data. Note that this concern applies to all the choices for the final statistical model in the paper.
Besides, my understanding is that the statistical model used in Experiment 2 is similar to the one used in Experiment 1. If it is not the case, this should be stated clearly. While fitting a linear mixed effects model is adapted to the data in Experiment 1 (modelExpt1 <- lmer( Liking ~ …) , this is not adapted for Experiment 2. In the second experiment, the outcome is binary (‘curved_preference_choice’ = 0, 1 or ‘sharp_preference_choice’ = 1, 0 to use the name of the variable in the spreadsheet). The inference process should therefore use a generalized mixed effects model and a binomial law (modelExpt2 <- glmer(curved_preference_choice~ …, family = binomial, …)). The remaining part of the statistical inference can then be similar to the one in Experiment 1.
Another point that could improve the quality of the analysis is to report p-values with a greater precision. Two significative numbers (in which case a true p = 0.001 should read p = 0.0010) would be a minimum.
In the last part of the analysis of Experiment 2, I don’t think that introducing another way to estimate the means in the different conditions adds anything. Mixed effects models provide much more information than this.
As a minor note, I think that the p-value for ‘Contour’ in Supplementary Table S1 should read < .001 ***, not > .001 ***.
Finally, as I have appreciated the authors’ good practice to add confidence intervals for the means, I would like to mention that adding confidence intervals for the Spearman-rank correlation is also important, even if not common yet (I am therefore not saying that this should be done here, that’s just a suggestion). A simple bootstrap procedure to do so can be easily implemented in R. I have used the papers and tutorials linked to this https://psyarxiv.com/h8ft7/to learn how.

Paradigm and methods -
The experimental paradigm chosen by the authors to assess the interaction between familiarity and preference for curvature is sound. A potential limitation to the conclusions is that the study uses self-reported familiarity. However, the authors clearly identify this limitation. Besides, I agree with the authors that repeated exposure to novel, unknown objects may rub out any effect of preference for curvature. I only have minor points to comment. I list them below.
l.173: The database includes almost 800 stimuli. What was the criterion to choose the 90 that were chosen?
l.180: If I am correct, the frame was 600 x 600 pixels and the stimuli were 300 x 300 pixels? I find the last part of the sentence hard to follow.
l.189, 196 (but also in Experiment 2): The task had 8 practice trials. With which stimuli, 8 from the set of 90, or other stimuli of the database? This is important as showing twice some of the stimuli may bias familiarity judgements.
l.188: ‘Each stimulus was presented on the centre of the screen until the participant responded on the sliding bar using the mouse.’ As written an inattentive reader may understand that the stimulus disappears once the mouse is moved. Consider changing for ‘Each stimulus was presented on the centre of the screen until the participant ***had*** responded on the sliding bar using the mouse.’
l.342: ‘In the 2AFC task, each pair of drawings was presented on the screen until participants responded as in previous studies (Munar et al., 2015; Corradi et al., 2018)’. Please insert comma after ‘responded’, otherwise the sentence implies that participants had to give the same answers as the ones that were given in previous studies to move on.
l.360 and following: the task in Experiment 2 is not explicit. We are told the words the instructions avoided but not what the instructions were. The paragraph ‘procedure’ for Experiment 2 should be as clear and explicit as the corresponding paragraph for Experiment 1, otherwise we don’t know what the participants were asked to do.

Writing -
The manuscript is generally clear even if some errors should be corrected and some explanation may be improved.
The main problem is in the statement of the conclusion given on lines 599-600. I f I am not misunderstood, when both objects are judged as equally familiar there is no clear preference in Experiment 2. The general conclusion as stated here, and in the abstract, is therefore abusive.
I list several other points below, following the order of the manuscript, not in order of importance.
Abstract: ‘unconventional participants’ is a bit odd; could you rephrase?
l.45: ‘using art-related stimuli’ -> ‘using non-representational art-related stimuli’ as art-related stimuli can be representational.
l.47: what do you mean by ‘shared preferences’?
l.133: ‘In this study, we examined preference for contour using drawings of common-use objects in two experiments’. Better ‘In this study, we examined preference for contour in two experiments using drawings of common-use objects’.
l.154: ‘or only model it’. The meaning of this part of the sentence, and hence of the full sentence, is not clear.
l.257: performed -> considered.
l.264: ‘as the influence of Contour’ -> ‘as assessing the influence of contour’.
l.281: ‘was our main objective’. As for line 264, should be rephrased.
l.321 and following: this interim discussion may be improved by explaining that the data suggest and interaction between familiarity and curvature to shape preference.
l.356: Both naïve and naïve are correct. The text uses both. Please choose one.
l.500: ‘Experiment 2, using the same stimuli and different participants, consisted of a 2AFC task simulating approach/avoidance responses’. Probably better: Experiment 2 used the same stimuli and different participants, and consisted of a 2AFC task simulating approach/avoidance responses’.
l.510: ‘Similarly, Corradi et al. (2018) found that the effect of preference for curvature decreased as the presentation time increased using the same task and stimuli.’ I suggest: ‘Similarly, and using the same task and stimuli, Corradi et al. (2018) found that the effect of preference for curvature decreased as the presentation time increased.’
I don’t understand the lines 576 to 584. In particular, the authors use ‘endorse’ with a meaning I am not familiar with.
l.602: ‘These findings support that…’ -> ‘These findings support the idea that…’.
l.608: ‘endorse’.
Misc.: several instances of ‘On another hand’ -> ‘On the other hand’.

---

## Round 0.2 · Minor Revisions

All three reviewers find your revisions to be acceptable and are ready to accept. I've selected "minor revisions" just to give you the opportunity to make a few small corrections before moving to the publishing stage.

In particular:
1) Reviewer 3 has made a number of additional suggestions that you will likely want to address.

2) For your comments in the discussion regarding the occurrence curved vs. straight lines in nature and human sensitivity to them, you might want to take a look at this article, as it highlights the issue I was raising (sensitivity to change in non-accidental vs. metric properties).
Amir, O., Biederman, I., & Hayworth, K. J. (2012). Sensitivity to nonaccidental properties across various shape dimensions. Vision Research, 62, 35–43.

3) Fig. 3B: change category label in middle panel to either “equal familiarity” or “equally familiar"

·

Basic reporting

No comment

Experimental design

No comment

Validity of the findings

No comment

Additional comments

Thank you for answering all my questions and adressig all my concerns.

Reviewer 2 ·

Basic reporting

no comment

Experimental design

no comment

Validity of the findings

no comment

Additional comments

The authors successfully responded to my previous comment.

Reviewer 3 ·

Basic reporting

I find that all the changes made by the authors have improved the manuscript and respond well to all the queries I had during the first phase of review. Both experiments are now clearly explained and fully reproducible.

Experimental design

No comment.

Validity of the findings

All the changes regarding the statistical inference and report have been satisfactorily implemented. The statistical modelling is now clear, coherent, and explicit.

Additional comments

I am happy with all the improvements made on the manuscript. They solve satisfactorily all the issues I found in the first submission. I therefore would like to congratulate the authors on this submission. I may have missed it, but I haven’t found any link or reference to the code used for the statistical inference in the manuscript. I think that it would be useful for the reader to have access to the osf entry created for this work. Please find below a short list of minor concerns in the updated manuscript.

Line 118: ‘e.g.’ instead of ‘i.e.’ in ‘for more realistic (i.e., photographs) or more abstract stimuli (i.e., irregular polygons).’?

Line 218: ‘Lastly, the apparent position of the objects in relation to the viewer was frontal view in 24 pairs, and ¾ view in 21 pairs.’ -> Lastly, the apparent position of the objects in relation to the viewer corresponded to a frontal view in 24 pairs, and to a ¾ view in 21 pairs.

Line 221: ‘Out of the computer-made drawings, 2 pairs were shaded, 13 pairs were not shaded, 8 pairs were in a ¾ view, and 7 pairs were in a frontal view.’ -> I think that ‘in ¾ view’ and ‘in frontal view’ are more correct, please check.

Line 333: ‘The familiarity selection task showed that the curved shapes were selected as the most familiar ones with an average of .49,…‘ ‘In a proportion’ or similar instead of ‘average’?

Line 491: same as line 333.

Line 507: please spell out ‘OR’ the first time it is used.

Line 562: ‘with 1499 samples suggested for a test at the 0.5 and 0.1 level’. Do you mean the 0.05 and 0.01 levels.

Lines 665 and 667: ‘In particular, Pepperell (2012) indicated that he uses a curvilinear perspective, instead of a linear perspective, because it is closer to the phenomenology of the perceiver.’ and ‘Straight lines in nature can appear curved in perception, a phenomenon that artists have shown (Pepperell & Haertel, 2014).’

I am struggling a bit with the thread here. Isn’t the statement more general than the one given in the first sentence (‘he uses’), and could both sentences be merged? For example, something such as ‘In particular, Pepperell (Pepperell (2012), Pepperell & Haertel, 2014) has shown that artists use curvilinear perspective instead of linear perspective because it is closer to the phenomenology of perception.’

---

## Round 0.3 · accepted · Accept

Thank you for addressing the remainder of the concerns.